# Regulation of Mitochondria-Associated Membranes (MAMs) by NO/sGC/PKG Participates in the Control of Hepatic Insulin Response

**DOI:** 10.3390/cells8111319

**Published:** 2019-10-25

**Authors:** Arthur Bassot, Marie-Agnès Chauvin, Nadia Bendridi, Jingwei Ji-Cao, Guillaume Vial, Léa Monnier, Birke Bartosch, Anaïs Alves, Cécile Cottet-Rousselle, Yves Gouriou, Jennifer Rieusset, Béatrice Morio

**Affiliations:** 1CarMeN Laboratory, INSERM U1060, INRA U1397, University Lyon 1, 69008 Lyon, France; arthur.bassot@univ-lyon1.fr (A.B.); marie-agnes.berger@inserm.fr (M.-A.C.); nadia.bendridi@inserm.fr (N.B.); jingwei.ji-cao@inserm.fr (J.J.-C.); anais.alves@etu.univ-lyon1.fr (A.A.); yves.gouriou@univ-lyon1.fr (Y.G.); jennifer.rieusset@univ-lyon1.fr (J.R.); 2HP2 Laboratory, INSERM U 1042, University Grenoble Alpes, 38000 Grenoble, France; guillaume.vial@inserm.fr; 3Cancer Research Centre (CRCL), INSERM U1052, 69008 Lyon, France; lea.monnier@inserm.fr (L.M.); birke.bartosch@inserm.fr (B.B.); 4Laboratory of Fundamental and Applied Bioenergetics (LBFA), INSERM U 1055, University Grenoble Alpes, 38000 Grenoble, France; cecile.cottet@univ-grenoble-alpes.fr

**Keywords:** mitochondria-associated endoplasmic reticulum membranes, nitric oxide, cyclic guanosine monophosphate (cGMP)-dependent protein kinase, hepatic glucose metabolism, metabolic flexibility

## Abstract

Under physiological conditions, nitric oxide (NO) produced by the endothelial NO synthase (eNOS) upregulates hepatic insulin sensitivity. Recently, contact sites between the endoplasmic reticulum and mitochondria named mitochondria-associated membranes (MAMs) emerged as a crucial hub for insulin signaling in the liver. As mitochondria are targets of NO, we explored whether NO regulates hepatic insulin sensitivity by targeting MAMs. In Huh7 cells, primary rat hepatocytes and mouse livers, enhancing NO concentration increased MAMs, whereas inhibiting eNOS decreased them. In vitro, those effects were prevented by inhibiting protein kinase G (PKG) and mimicked by activating soluble guanylate cyclase (sGC) and PKG. In agreement with the regulation of MAMs, increasing NO concentration improved insulin signaling, both in vitro and in vivo, while eNOS inhibition disrupted this response. Finally, inhibition of insulin signaling by wortmannin did not affect the impact of NO on MAMs, while experimental MAM disruption, using either targeted silencing of cyclophilin D or the overexpression of the organelle spacer fetal and adult testis-expressed 1 (FATE-1), significantly blunted the effects of NO on both MAMs and insulin response. Therefore, under physiological conditions, NO participates to the regulation of MAM integrity through the sGC/PKG pathway and concomitantly improves hepatic insulin sensitivity. Altogether, our data suggest that the induction of MAMs participate in the impact of NO on hepatocyte insulin response.

## 1. Introduction

Within the hepatocyte, mitochondria and the endoplasmic reticulum (ER) communicate through contact points, called mitochondria-associated ER membranes (MAMs), in order to exchange phospholipids and calcium and regulate cellular functions [1]. Although the exact composition of MAMs is still unclear, key proteins participating in the MAM structure and function have been identified, such as inositol 1,4,5-trisphosphate receptor (IP3R), voltage dependent anion channel (VDAC), glucose-regulated protein 75 (Grp75), cyclophilin D (CypD), and mitofusin 2 (Mfn 2) [2]. Beyond composition, the physiological regulation of MAMs is poorly understood. We recently demonstrated that MAMs are major nutrient sensors in the liver, as they adapt to glucose availability in order to regulate mitochondrial dynamics and bioenergetics [3]. Furthermore, MAMs were involved in the control of hepatic insulin signaling [2] and ER–mitochondria miscommunication was associated with hepatic insulin resistance, but with conflicting results [2,4,5]. Consequently, MAMs are now presented as subcellular signaling platforms important for hepatic metabolism. Thus, understanding key mechanisms regulating MAM integrity in the liver could help identify new strategies to modulate hepatic insulin sensitivity, which is a key lever to prevent metabolic disorders associated with obesity [6,7].

Nitric oxide (NO) is a gaseous signaling molecule whose fundamental role in the organism’s homeostasis was discovered in the 1990s [8]. Since then, NO has been involved in many physiological and pathological processes, especially at the level of the liver [8]. Notably, mice knockout for the endothelial isoform of the NO synthase (eNOS) were shown to develop hepatic insulin resistance [9]. eNOS is the main NOS producing NO in the liver under physiological conditions [10]. eNOS activity results in pulsatile release of NO over a period of seconds and minutes [11]. It is important to contrast the physiological effects of eNOS with those observed when the inducible isoform of NOS (iNOS) is stimulated. Under physiological conditions, iNOS is expressed at very low concentrations in the liver [10], but its transcription can be greatly increased in response to inflammatory and oxidative stresses [12]. Once induced, iNOS activity chronically produces NO for hours or days [12].

The impact of NO on cell function is mediated by two main pathways. The first one is activated by nanomolar concentrations of NO in physiological conditions [13]. It involves the intracellular production of cyclic guanosine monophosphate (cGMP) from guanosine triphosphate (GTP) following the binding of NO to the subunits of the cytosolic isoform of guanylate cyclase (sGC). cGMP activates the cGMP-dependent protein kinase G (PKG), which is present in hepatocytes under two isoforms, PKG type I alpha and beta [14]. The second pathway is activated by a wide range of NO concentrations, from nanomolar to millimolar, and involves S-nitrosylation reactions [15]. The impacts of these two pathways on a particular mechanism or target are sometimes complementary but often opposite, conferring onto NO a complex role as a double-edged messenger [16].

NO generated by eNOS was shown to regulate the insulin signaling pathway in the liver [9]. Mechanisms involve both phosphorylation and S-nitrosylation of the two enzymes phosphatase and tensin homolog (PTEN) and protein kinase B (also known as Akt) [17,18]. In addition, eNOS-derived NO is well known to tightly regulate mitochondrial functioning. In physiological concentrations, NO regulates mitochondrial network fusion by phosphorylating and inhibiting dynamin-related GTPase (DRP1) through the sGC/PKG pathway [19]. NO also stimulates mitochondrial biogenesis through the sGC/PKG pathway in several tissues including the liver [20]. As the contact sites between mitochondria and the ER (MAMs) have emerged as a crucial hub for insulin signaling in the liver, we hypothesize that NO generated by eNOS could regulate hepatic insulin sensitivity by targeting MAMs.

Therefore, our objective was (1) to examine whether NO regulates the integrity of MAMs in the liver under physiological conditions, (2) decipher the pathways involved, and (3) understand whether regulation of MAMs is involved in the impact of NO on insulin signaling. Our results evidence that, under physiological conditions, NO participates to the regulation of MAM integrity through the sGC/PKG pathway and that MAM integrity could play a key role in mediating the impact of NO on hepatocyte insulin response.

## 2. Materials and Methods

### 2.1. Chemicals

Primary antibodies for AKT (total, rabbit), phospho AKT (Ser473, rabbit), glycogen synthase kinase 3 beta (GSK3β; total, rabbit), phospho GSK3β (Ser9, rabbit), and green fluorescent protein (GFP, rabbit) were from Cell Signaling Technology (Danvers, MA, USA). Those for, IP3R1 (rabbit), fetal and adult testis-expressed 1 (FATE-1; mouse), eNOS (NOS3, mouse), and iNOS (NOS2, mouse) were from Santa Cruz Biotechnology (Dallas, TX, USA). The antibodies for CypD (cyclophilin F, mouse) and VDAC1 (mouse) were from Abcam (Paris, France) and the one for tubulin α (mouse) was from Sigma-Aldrich (Saint-Quentin Fallavier, France). Duolink^®^ proximity ligation assay and Trizol Reagent kits were from Sigma-Aldrich (Saint-Quentin Fallavier, France). NucleoSpin^®^ DNA RapidLyse kit was from Macherey-Nagel (Hoerdt, France). PrimeScript reagents RT kit (Takara Bio Inc., USA). Bradford protein assay kit was from ThermoFisher Scientific (Courtaboeuf, France). Other chemicals were from Sigma-Aldrich (Saint-Quentin Fallavier, France), Qiagen (Courtaboeuf, France), ThermoFisher Scientific (Courtaboeuf, France), or Cell Signaling Technology (Danvers, MA, USA).

### 2.2. Animal Studies

Animal studies were performed in accordance with the French guidelines for the care and use of animals [21] and were approved by a regional ethic committee (CECCAP LS_2017_004). Male C57BL/6JOlaHsd mice (5-weeks-old) (Envigo, France) were housed in groups of 5 on sawdust bedding in plastic cages with enriched environment up to the age of 5 months. Artificial lighting was provided on a fixed 12 h light–dark cycle with standard food and water ad libitum. Placebo (saline), l-Name (ω-nitro-l-arginine methyl ester hydrochloride; 25 mg/kg), BH4 (tetrahydrobiopterin; 12.5 mg/kg), and both BH4 (12.5 mg/kg) and l-Name (25 mg/kg) treatments were given by intraperitoneal injection the day before (morning and evening) and 2 h before tests. Overnight fasted mice received an intraperitoneal injection of either saline (*n* = 5 to 7 mice per treatment) or insulin (0.75 U/kg, *n* = 5 mice per treatment) 15 min before euthanasia. Thereafter, the liver was quickly removed, weighted, and divided for further analyses. Hepatic glucose production was assessed in overnight fasted mice following intraperitoneal injection of pyruvate (2 g/kg, *n* = 8 mice per treatment). Blood glucose was measured before and 15, 30, 45, 60, 90, and 120 min post pyruvate injection in a drop of blood from the tail using a glucometer (Accu-Check Performa^®^, Roche Diabetes Care, Meylan, France).

### 2.3. Cell Culture

Experiments were conducted on hepatocellular carcinoma Huh7 cells [22] and primary rat hepatocytes. Primary rat hepatocytes were isolated via a modified collagenase perfusion method, as described previously [23,24]. Huh7 cells and primary rat hepatocytes were cultured in Dulbecco’s Modified Eagle’s Medium (DMEM, PAA Laboratories) containing 5.5 mM and 16.5 mM glucose, respectively, at 37 °C and in a humid atmosphere with 5% CO_2_. DMEM was supplemented with 2 mM glutamine, 2 mM antibiotic/antimycotic, and 10% fetal calf serum (FCS), constituting the standard mediums. Huh7 cells in exponential growth (#80% confluent) were used for all experiments. In all experiments, cells were first cultured for 24 h in a standard medium before any treatment started. When GFP (used as a control) and FATE-1-GFP (fetal and adult testis-expressed 1) adenoviruses were used [25], the medium (DMEM containing 5.5 mM glucose, 2 mM glutamine, 2 mM antibiotic/antimycotic, and 10% FSC) was changed 16 h after infection and experimentations were performed 48 h later. Doses of adenoviruses were calculated so that the viral infection is greater than 80% of the cells per well (checked by microscopy). This corresponds to 25 times higher expression of GFP/TBP (TATA-box binding protein) mRNA measured by PCR compared to the control without infection (after RNase-Free DNAse set Qiagen treatment of each sample). Alternatively, we used Ad-cherry and Ad-FATE-1 (non-GFP) adenoviruses when we measured NO production as GFP interferes with the Daf-FM (3-Amino-4-(*N*-methylamino)-2’,7’-difluorofluorescein) signal.

#### 2.3.1. Generation of Clustered Regularly Interspaced Short Palindromic Repeats (CRISPR)/Cas9 Constructs for CypD Knockout and Establishment of Cell Lines 

sgRNAs targeting cyclophilin D (exon 1: 5′TGACGTCGCGCGCGCCCAGA3′, exon 6: 5′GATCTTTACTGAGCTCGCCG3′) were cloned into lentiCRISPRv2 (Cat No. 52961, Addgene, Cambridge, MA) as previously described [26]. The resulting plasmids (8.1 µg) were each packaged into lentiviruses by cotransfection with an HIV Gag-Pol packaging construct (8.1 µg), and a VSV-G expression construct (2.7 µg). DNAs were transfected into 2.5 × 10^6^ HEK293T cells (ATCC CRL-1573) seeded the day before in 10-cm plates using a calcium phosphate based protocol (CLONTECH Laboratories, Inc. Mountain View, CA, USA), as described previously [27]. The medium (8 mL/plate) was replaced 16 h after transfection. The virus-containing supernatants were harvested 24 h later, filtered through 0.45-µm pore-sized membranes, and used immediately for transduction. Then, ~1.2 × 10^6^ Huh7 cells in 10-cm plates were transduced with lentiviruses carrying exon 1 and exon 6 specific sgRNAs at a multiplicity of infection (MOI) below 1. Three days later puromycin was added (final concentration 10 μg/mL) to select for sgRNA/Cas9-positive cells. The expanded cell culture was then subjected to single cell sorting using FACS (fluorescence-activated cell sorting; BD FACSAria III SORP) to obtain monoclonal cell lines. Genomic DNA was then extracted from each clone for PCR amplification and validation of the sgRNA/Cas9 excision site (exon1 f 5′AACCTGGGCAAGCCAATAAAGG3′; r 5′GGAAACTGAGGCCCAGAGCTC3′; exon6 f 5′ AACATGGATTTGTGTTCACCTT3′; r 5′TGGGTGCTAAGGTCGTTTGTCT3′). CypD-knockout (KO) was finally confirmed by Western blotting and RT-qPCR (f 5′AAAGACAGACTGGTTGGATGGC3′; r 5′GCACCCTGGCCACAGATTAG3′).

#### 2.3.2. Modulation of NO Concentrations and sGC/PKG Pathway

In all in vitro experiments, 24 h after seeding, cells were treated for 24 h with either arginine (1 mM, the eNOS substrate), l-Name (1 mM, an inhibitor of eNOS), or NONOate (diethylamine NONOate sodium salt hydrate; 1 mM, a NO donor) to modulate NO concentrations, and with either BAY41-2272 (3-(4-amino-5-cyclopropylpyrimidin-2-yl)-1-(2-fluorobenzyl)-1*H*-pyrazolo [3–b]pyridine; 2 µM, an activator of sGC), 8-pCPT-cGMP (8-(4-chlorophenylthio)-guanosine 3′,5′-cyclic monophosphate sodium salt; 100 µM, an activator of PKG), or KT-5823 (1 µM, an inhibitor of PKG) to modulate the sGC/cGMP/PKG pathway. Treatments were added to the standard medium to maintain cell viability. Control conditions received the same amount of dimethyl sulfoxide (DMSO) as the treated conditions.

#### 2.3.3. Insulin and Wortmannin Stimulation

Twenty-one hours after treatment initiation, mediums were FBS deprived for 3 h. To inhibit the insulin signaling pathway, cells were treated with wortmannin (1 µM) during the last hour of FBS deprivation. Then, 100 nM of human insulin was added for 15 min. The reaction was stopped on ice and cell lysates were collected in Ripa buffer (1% sodium deoxycholate, 0.1% sodium dodecyl sulfate (SDS), 1% NP-40 permeating solution, in PBS supplemented just before the extraction with 1 mM dithiothreitol (DTT), 5 mM ethylenediaminetetraacetic acid (EDTA), 20 mM sodium fluoride (NaF), 1 mM sodium orthovanadate (Na_3_VO_4_), and protease inhibitor 1× (1/1000)). 

### 2.4. Assessment of NO Concentrations

On Huh7 cells (#80% confluent), NO concentrations were measured in a glass-bottom 96-well plate (Greiner Bio One, Courtabeuf, France) using Daf-FM diacetate probe (15 µM in PBS, 30 min at 37 °C). Cell fluorescence was measured using Fluoroskan Ascent (Thermo Electron Corporation^®,^ Illkirch, France) (excitation 485 nm/emission 538 nm) after several washings with PBS. NO concentrations were similarly assessed on supernatants of fresh liver homogenates (15 µM Daf-FM diacetate probe, 30 min at room temperature).

### 2.5. Isolation of MAMs

MAM fractions were isolated from freshly isolated livers by differential ultracentrifugation using a Percoll gradient (225 mM Mannitol, 25 mM HEPES (4-(2-hydroxyethyl)-1-piperazineethanesulfonic acid), 1 mM EGTA (ethylene glycol-bis(β-aminoethyl ether)-*N*,*N*,*N*′,*N*′-tetraacetic acid), and 30% Percoll) according to Wieckowski et al. [28] and as described by Tubbs et al. [2]. The various steps of centrifugation made enabled recovering pellets made of ER, pure mitochondrial, and pure MAM fractions. All fractions were suspended in Ripa buffer before being stored at −80 °C for further analyses.

### 2.6. Transmission Electronic Microscopy

Transmission electronic microscopy protocol of liver fixation, sections, and image acquisition were performed according to Bonnard et al. [29]. Pictures of liver sections were taken at 30,000× magnification. The quantitative analysis was performed using a morphometric application programmed on ImageJ^®^ (Fiji) as described by Theurey et al. [3]. This application determines the mitochondrial circumference, ER length, ER and mitochondria contact length, and the distance between the two organelles (from 0 to 50 nm). Results were normalized to mitochondria circumference and expressed in percentage [3]. A minimum of 10 images were taken per sample, and five mice were analyzed per group.

### 2.7. Fluorescence Imaging 

#### 2.7.1. In Situ Proximity Ligation Assay (PLA)

After treatment as described above, fixed and permeabilized cells (#5 × 10^4^ in glass-bottom 35 mm cells [2]) were incubated overnight at 4 °C with a binary mixture of anti-IP3R1 (1/500 dilution) and VDAC1 (1/500 dilution) primary antibodies. Thereafter, after two washes with TBS-Tween (tris-buffered saline polysorbate 20) 0.05%, cells were incubated (1 h at 37 °C) with the complementary secondary antibody (rabbit PLUS and mouse MINUS). The proximity ligations were then performed according to the manufacturer’s protocol. Preparations were mounted in Duolink II mounting medium containing DAPI (4′,6-diamidino-2-phenylindole). Fluorescence was analyzed with a Zeiss inversed fluorescent microscope using the AxioVision program. Fluorescence dot signals were quantified using BlobFinder software (v3.2, center for Image Analysis, Uppsala University). Results were expressed as number of blobs per nucleus. A minimum of 10 images were taken per sample, and three independent series were performed for each treatment.

#### 2.7.2. MitoTracker Green^®^

Morphology of the mitochondrial network was explored using the fluorescent probe MitoTracker Green (0.5 μM, 30 min at 37 °C). After treatment as described above, live cells were washed several times with PBS and maintained in PBS for image acquisition using a Zeiss inversed fluorescent microscope as described above. The quantitative analysis of mitochondrial morphology was performed using a morphometric application programmed on ImageJ^®^ (National Institutes of Health, MA, USA) [3]. This application determines the mean area, perimeter, major, and minor radiuses of cell mitochondria. Mitochondrial fusion/fission was evaluated using the form factor (FF = 4π x area/perimeter^2^) and the aspect ratio (AR = major radius/minor radius) [30] after top hat filtering was applied to the raw images to remove noise and to obtain a precise definition of the mitochondrial morphology. A minimum of 10 images were analyzed per sample, and three independent series were performed for each treatment.

### 2.8. Western Blot

Proteins from cells or mice liver were extracted in Ripa buffer. Protein concentrations were assessed using the Bradford protein assay kit (ThermoFisher Scientific). Laemmli buffer 6× solution (Tris HCL 150 mM pH 7.4, glycerol 36%, SDS 10%, β-Mercaptoethanol, Bromphenol blue 0.03%) was added to 15 µg of protein. After thermal denaturation (10 min at 75 °C), samples were deposited in acrylamide/bis-acrylamide 10% gels. Western blots were then conducted as described previously [2,24]. The signal was quantified using the ImageLab software (Bio-Rad, Marnes la Coquette, France). Results were expressed in relation to total protein for a phosphorylated form and to tubulin for total protein.

### 2.9. Gene Expression

#### 2.9.1. mRNA Extraction and Reverse Transcription

Total RNA from Huh7 cells was extracted using Trizol Reagent kit following the method of Chomczynski and Sacchi [31]. Cells infected with an adenovirus (see above) were treated with DNAse I (RNase-Free DNase Set) and extracted a second time. RNA concentration was measured using the Nanodrop 2000 (ThermoFisher Scientific) according to Thermo scientific recommendations. Reverse transcription (RT) of RNA to complementary DNA (cDNA) was performed using 1 µg of total RNA in 10 µL of total reaction volume of PrimeScript reagents RT kit. Total RNA samples were treated with DNase to prevent the potential contamination with genomic DNA. Primers are listed in Appendix A.

#### 2.9.2. DNA Extraction (for Mitochondrial DNA Analysis)

Genomics DNA of Huh7 cells were extracted using the nucleoSpin^®^ DNA RapidLyse kit. DNA concentration was measured using the Nanodrop 2000 (ThermoFisher Scientific).

#### 2.9.3. Measurement of Gene Expression by PCR

Gene expression of proteins COX-1 (cyclooxygenase-1; a mitochondrial gene), PPIA (peptidylprolyl isomerase A; a nuclear gene), and GFP was explored by qPCR (quantitative real-time polymerase chain reaction) using SYBR^®^ Green mix and Rotor-Gene 6200 (Corbett Research Cambridge, UK). Each sample assay was performed in duplicate. Relative expression of the different genes was estimated by the cycle threshold (Ct) and values were normalized using the TATA-binding protein (TBP) mRNA as a housekeeping gene. Primers are listed in Appendix A.

### 2.10. Oxygraphy

Oxygen consumption was monitored using Mitocell MT200 with a 782 Oxygen meter (Strathkelvin Instrument Ltd., Newhouse, Scotland) in intact Huh7 cells and primary rat hepatocytes in DMEM medium with glucose and treatments. Prior to oxygraphy measurements, cells were plated and treated as described above. Basal respiration was assessed at 37 °C with treatments. The addition of 10 µg/mL of oligomycin provided uncoupled (non-phosphorylating) oxygen consumption.

### 2.11. Statistical Analysis

All data are presented as means ± standard error of the mean (SEM). After verification of data normality (with Shapiro–Wilk or Agostino and Pearson omnibus normality test) and homogeneity (with Bartlett’s test), one-way or two-way (when appropriate) analyses of variance (ANOVA) were performed to test the effect of the experimental conditions. When a significant effect was detected, a posteriori Bonferroni correction test was used to analyze pairwise differences. Dunnett’s test was performed to compare the difference to the control only. For all other comparisons, data were analyzed with Kruskal–Wallis or Student’s t-test. Statistical analysis was performed using StatView^®^ (SAS, Cary, NC, USA) and GraphPad Prism^®^ software (GraphPad Software, San Diego, CA, USA). *p* < 0.05 was considered significant. 

## 3. Results

### 3.1. NO Regulates MAM Integrity and Insulin Response in the Liver in Vivo

In normal conditions, NO is produced in hepatocytes by the endothelial isoform of the NO synthase (eNOS) [10]. We confirmed by Western blotting that eNOS was the main isoform expressed in the liver of male C57BL/6JOlaHsd mice, and that iNOS was not significantly expressed (Appendix A). NO production (Figure 1B) was then manipulated accordingly using both stimulatory or inhibitory approaches described in Figure 1A. ER–mitochondria interactions in the liver of mice were investigated using subcellular fractionation and TEM analysis. Purity of subcellular fractions was validated by Western blot analysis of specific protein markers (Appendix A). Enhancing NO production using BH4 (12.5 mg/kg), a cofactor for eNOS, increased 45% and 65% the amount of proteins in the MAM fraction expressed in relation to total and mitochondrial proteins, respectively (*p* < 0.05, Figure 1C,D). In contrast, decreasing NO production using l-Name (25 mg/kg), an eNOS inhibitor, alone or combined to BH4 decreased about 40% of the amount of proteins in the MAM fraction (*p* < 0.05, Figure 1C,D). More specifically, the impact of l-Name alone on the MAM fraction expressed relative to mitochondrial proteins was not significant (Figure 1D) since the treatment also induced a 33% decrease in the amount of mitochondrial proteins, suggesting a decrease in mitochondrial density (Figure 1E). In agreement, based on the quantifications made from TEM images, BH4 treatment increased the percentage of interaction between ER and mitochondria (expressed in percent of mitochondrial circumference) for inter-organite distances ranging from 30 to 50 nm (*p* < 0.05, right panel Figure 1F). On the reverse, treatment with l-Name alone or in combination to BH4 induced a 40% decrease in all spacing ranges in the percentage of interaction between ER and mitochondria (Figure 1F). These changes in MAM integrity following treatments were observed without any significant modification of the expression of some key MAM proteins (Mfn2, Grp75, CypD, VDAC1, Opa1) in hepatic MAM fractions of treated mice (Appendix A).

We then investigated whether NO also impacted hepatic insulin action. Liver insulin response, assessed through Akt phosphorylation at Ser473 (Figure 2A,B), was altered in a comparable way to MAMs in response to treatments. Briefly, BH4 increased the insulin-stimulated phosphorylation of Akt by 32% (*p* < 0.05, Figure 2B). Decreasing NO production using l-Name alone or combined with BH4 decreased the insulin-stimulated phosphorylation of Akt by 40% and 70%, respectively (*p* < 0.05, Figure 2B). Similar results were obtained when assessing insulin-stimulated phosphorylation of GSK3β (Figure 2A,C). Some insulin signaling proteins, such as Akt and GSK-3β, are located at MAMs [2] and may be involved in mediating the impact of NO on insulin response. We thus explored whether NO could alter its expression or phosphorylation state in hepatic MAM fractions from treated mice. However, no significant difference was observed between treatments (Appendix A). Lastly, to confirm the impact of modulating NO on liver metabolism, in vivo pyruvate tolerance tests were performed. BH4 alone did not significantly improve the glycemic response, although glycemia was most often below that of control animals (Figure 2D). In contrast, glycemia was significantly higher from 60 to 120 min after pyruvate injection in mice treated with l-Name alone or combined with BH4, supporting the development of hepatic insulin resistance following acute l-Name treatment (*p* < 0.05, Figure 2D). These experiments highlight that NO upregulates both MAM integrity and insulin response in the liver in vivo. Mechanisms were then explored in vitro on human Huh7 cells and primary rat hepatocytes. 

### 3.2. NO Regulates MAM Integrity and Insulin Response in Hepatocytes In Vitro

We first investigated whether human Huh7 cells were a relevant model to investigate physiological responses to NO. Western blotting (Appendix A) and immunofluorescence (Appendix A) evidenced that eNOS was the main NOS producing NO in Huh7 cells, and that iNOS was not significantly expressed. To investigate MAM integrity in vitro, we used in situ proximity ligation assay (PLA) targeting the proximity between the mitochondrial channel VDAC1 and the ER receptor IP3R1 at the MAM interface [32]. The method detects the physical proximity between two proteins close to 40 nm, which was compatible with ER–mitochondria contact sites and with our observations in vivo by TEM. Previous work by our team also showed consistent results between quantifications of ER–mitochondria interactions made from TEM and VDAC1/IP3R1 in situ PLA [3]. NO production (Figure 3B) was manipulated using strategies illustrated in Figure 3A. Given that NO production from eNOS is subtle and intermittent, we first performed a kinetic study to access the impact of acute arginine addition and withdrawal on ER–mitochondria interactions. Interestingly, our results indicated that the regulation of MAMs by NO is dynamic, as NO upregulates MAM integrity as soon as 5 min of treatment, and this effect is rapidly reversible (Appendix A). However, the effect of NO on MAMs is persistent as it is also present after 24 h of treatment, and we chose this time to analyze the repercussions of NO modulation on MAMs, mitochondria, and insulin signaling in Huh7 cells.

We found that enhancing NO concentration using arginine 1 mM, the substrate of eNOS, or NONOate 1 mM, a NO donor, for 24 h increased by 1.4–2.5 fold the number of VDAC1-IP3R1 dots per cell in Huh7 cells (*p* < 0.05, Figure 3C). Similar results were obtained in primary rat hepatocytes (*p* < 0.05, Figure 3D). In contrast, decreasing NO concentration by l-Name treatment reduced 40% on average, both basal and arginine-stimulated levels of VDAC1-IP3R1 proximity in Huh7 cell (*p* < 0.05, Figure 3C), and similar results were obtained in primary rat hepatocytes (*p* < 0.05, Figure 3D). In these conditions, neither the expression of VDAC1 nor the expression of IP3R1 was modified by treatments (Appendix A). 

We also verified that NO altered mitochondrial network architecture, density, and respiratory activity, in agreement with the literature [33]. Enhancing NO concentrations using either arginine or NONOate for 24 h increased mitochondrial network fusion, as AR and/or FF were increased (*p* < 0.05, Figure 3E). In contrast, these treatments did not significantly alter the cellular mtDNA content (Figure 3F) and oxygen consumption (Figure 3G). On the reverse, decreasing NO concentrations using l-Name 1 mM caused mitochondrial network fission (*p* < 0.05, Figure 3E) and reduced the cellular mtDNA content (*p* < 0.05, Figure 3F) and oxygen consumption (*p* < 0.05, Figure 3G). Similar results were obtained on primary rat hepatocytes, as illustrated for cell oxygen consumption in Appendix A. Finally, we found no effect of NO modulation on cell viability (Appendix A) and on the expression of caspase 3 and ER stress markers, excluding any effects on mitochondrial apoptosis (Appendix A) and on ER stress (Appendix A).

We next investigated the effects of NO modulation on insulin signaling, by measuring Akt phosphorylation at Ser473. Similar to results on MAMs, increasing NO concentration for 24 h increased insulin-mediated Akt phosphorylation at Ser473, in both Huh7 cells (*p* < 0.05, Figure 4A) and primary rat hepatocytes (*p* < 0.05, Figure 4B). Decreasing NO concentration using l-Name reduced the response of Akt to insulin (Figure 4A,B), but the level of significance was reached only in primary rat hepatocytes (*p* < 0.05, Figure 4B). Altogether, these results further support that NO simultaneously regulates MAM integrity and insulin response in hepatocytes.

### 3.3. NO Regulates MAM Integrity and Insulin Response in Hepatocytes Through the sGC/PKG Pathway 

NO is a well-known activator of the sGC/PKG pathway [13] (Figure 5A), we therefore hypothesized that this pathway may be involved in the effects of NO on both MAMs and insulin signaling. Using pharmacological strategies illustrated in Figure 5A, we explored in Huh7 cells the impact of modulating this pathway on MAM integrity using the in situ PLA technique. PKG inhibition using KT5823 1 µM for 24 h decreased by 20% the proximity between VDAC1 and IP3R1 compared to control cells (*p* < 0.05, Figure 5B), and prevented the positive impact of arginine and NONOate on MAM integrity (Figure 5B). On the reverse, inducing PKG using either 8-pCPT-cGMP 100 µM, a PKG activator, or BAY41-2272 2 µM, a sGC activator, for 24 h enhanced by 83% and 33% the number VDAC1/IP3R1 contact points per cell, respectively (*p* < 0.05, Figure 5C).

Impact of activating and inhibiting sGC and PKG on mitochondrial network architecture (Figure 5D), density (Figure 5E), and respiratory activity (Figure 5F) was also investigated. Only sGC activation by BAY41-2272 2 µM significantly altered the mitochondrial network architecture, inducing fusion (Figure 5D). Mitochondrial density was slightly decreased by KT5823 1 µM but was not significantly altered by other treatments (Figure 5E). By contrast, PKG inhibition using KT5823 1 µM reduced the cellular oxygen consumption by 25% compared to control cells, while its activation using 8-pCPT-cGMP 100 µM enhanced by 50% (*p* < 0.05, Figure 5F).

As for ER–mitochondria interactions, PKG inhibition using KT5823 1 µM decreased the insulin-stimulated Akt phosphorylation by 20% in control and arginine-treated Huh7 cells (*p* < 0.05, Figure 6A), while PKG activation using 8-pCPT-cGMP 100 µM or BAY41-2272 2 µM improved it by 75% on average (*p* < 0.05, Figure 6B). These results thus support that the NO/sGC/PKG pathway is involved in the regulation of MAM integrity and insulin response in hepatocytes.

### 3.4. The Insulin Signaling Pathway is Not Required in the Regulation of MAM Integrity by NO 

At this stage, we could not exclude that the effects of NO on MAMs could be secondary to the regulation of insulin action. To investigate this hypothesis, Huh7 cells were incubated for 24 h with activators of the sGC/PKG pathway (i.e., arginine 1 mM, NONOate 1 mM, or 8-pCPT-cGMP 100 µM) in the presence of wortmannin 1 µM, a PI3K/Akt inhibitor (Appendix A). As expected, wortmannin 1 µM fully prevented the insulin-induced Akt phosphorylation even in the presence of activators of the sGC/PKG pathway (*p* < 0.05, Figure 4C). Importantly, NO production was preserved under wortmannin 1 µM treatment (Appendix A) and the impact of activators of the sGC/PKG pathway on MAM integrity was maintained to a level similar to that observed without wortmannin (Figure 4D). Therefore, the regulation of MAM integrity by NO is not consecutive to the activation of the insulin signaling pathway, suggesting that NO-induced MAM integrity may play a role in regulating the hepatocyte insulin response.

### 3.5. MAM Integrity is Necessary for NO to Impact on Insulin Response In Vitro

Next, we investigated whether the integrity of close contact points between ER and mitochondria were essential for the action of NO on the hepatocyte insulin response. For that purpose, we used two different approaches based on an engineered disruption of MAM integrity in Huh7 cells. Firstly, we used Huh7 knockout cells for cyclophilin D (CypD) generated using CRISPR/Cas9 (Appendix A), as CypD has been recently involved in the regulation of MAM integrity and function [24,34]. In agreement, we confirmed in this study that loss of CypD disrupted ER–mitochondria interactions in CRISPR/Cas9 Huh7 cells (Figure 7A), as previously found in CypD-silenced Huh7 and hepatocytes from CypD-KO mice [24]. Importantly, this model evidenced that MAMs were not responding to the modulation of NO concentration (Figure 7A, Appendix A). Similar results were obtained on an independent CRISPR/Cas9 CypD-KO clone (Appendix A) and using CypD siRNA in standard Huh7 cells (Appendix A), even in the presence of the NO donor NONOate (Appendix A). Noteworthy, the impact of NO on insulin-induced phosphorylation of Akt was lost on those CypD-related models with MAM disruption (Figure 7C).

Secondly, we infected Huh7 cells with an adenovirus overexpressing a protein spacer, fetal and adult testis-expressed 1 (FATE-1) protein, which specifically disrupts contact points between ER and mitochondria [25,35]. We confirmed here that the overexpression of FATE-1 reduced VDAC1-IP3R1 interactions in Huh7 cells (Figure 7B). Importantly, this approach strengthened that neither MAMs (Figure 7B) nor insulin response (Figure 7D) were responding to the manipulation of NO concentration (Appendix A) when the integrity of MAMs was disrupted. Indeed, ER–mitochondrial contact points through IP3R1-VDAC interaction were no longer regulated by NO when MAM integrity was disrupted by FATE-1 (Figure 7B). In addition, the effect of NO on insulin signaling was lost when MAM integrity was altered through FATE-1 overexpression (Figure 7D). These combined data, obtained using two distinct strategies to disrupt MAMs, therefore support that the regulation of MAM integrity may play a key role in mediating the impact of NO on hepatocyte insulin response.

## 4. Discussion

Fifteen years ago, NO derived from iNOS was shown to induce combined alterations in mitochondria and ER, resulting in modified calcium fluxes between the two organelles [36]. However, the physiological significance of NO production by eNOS has remained unexplored to date. In line with this statement, our work evidences that physiological concentration of NO closely monitors the integrity of MAMs through the sGC/PKG pathway. More importantly, our study supports that this step could be crucial for the regulation of hepatic insulin signaling. Our work therefore highlights that the impact of NO on the dynamics of interaction between ER and mitochondria could be a key process in regulating the hepatic insulin signaling pathway.

The demonstration was conducted in vivo and in vitro under physiological conditions to specifically explore the importance of NO derived from eNOS. Hepatocytes constitutively express the eNOS isoform but can also express the inducible isoform iNOS [10]. These two enzymes demonstrate different modes of NO production as iNOS may generate up to 1000 times more NO than eNOS. This results in distinct and often opposite impacts on the hepatocyte biological functions [10]. This is why we used only lean mice in this study and not diabetic mice which highly express iNOS [37]. We further validated that our models mainly expressed the eNOS isoform (Appendix A) and that NO was produced in a physiological range (i.e., at concentrations that do not produce harmful metabolic effects). As measurements of NO production on tissues and cells were relative and could not inform about the absolute concentration values, we used concentrations that were a priori non-deleterious for mitochondrial function and insulin signaling. In agreement, we reproduced the known impacts of NO on mitochondrial network architecture, biogenesis, and respiration [19,20,38] and on insulin signaling [9].

Interestingly, our study evidenced for the first time that NO contributes to the upregulation of the interactions between ER and mitochondria, here named MAM integrity. This result was confirmed both in vitro and in vivo and was consistently observed whatever the technique used to quantify ER–mitochondria interactions (subcellular fractionation, TEM, and in situ PLA). The positive effect of NO on MAMs is dynamic, being present as soon as 5 min of treatment, and is maintained during 24-h treatment. We chose a 24-h treatment in all in vitro experiments to be consistent with the in vivo procedures. These conditions also guaranteed stable and reproducible readings, compatible with insulin response measurements.

Inhibition of eNOS by l-Name reduced both basal and NO-induced ER–mitochondria interactions. The decrease in MAM integrity observed when NO production was inhibited by l-Name may be partly related to a reduction in mitochondrial density, a feature well described in different cell types and tissues [20,39,40]. However, in vivo and in vitro markers of mitochondrial content showed that alterations in mitochondrial density were not the only determinants of the coupling between the two organelles. In vitro assessments further evidenced that alterations in ER–mitochondrial contacts were co-occurring with changes in mitochondrial dynamics. Consistent with our observations when NO production was suppressed by l-Name, a decrease in MAM integrity caused by hyperglycemia [41] or by silencing key proteins [42,43,44] has been shown to induce mitochondrial fission, although it was found that MAMs also provide an interface for mitochondrial fragmentation [45,46,47]. Our observations are further in agreement with the findings that enhancing NO production and activating the sGC/PKG pathway are necessary for the elongation of mitochondria [19]. Mechanisms involve the inhibitory phosphorylation of the mitochondrial fission factor DRP1 [19,48]. However, we did not investigate in the present study whether the modulation of MAM by NO is required to mediate the effects of NO on mitochondria dynamics. This interesting question requires further investigation.

It is well acknowledged that the sGC/PKG pathway mediates part of NO’s impact on cell biology. This pathway has been involved in the regulation by NO of mitochondrial biogenesis [20] and network architecture [19]. Our data evidence indicates that it is an important pathway in regulating MAM integrity. Indeed, activation and inhibition of key steps of the sGC/PKG pathway significantly impacted MAM integrity in vitro. Interestingly, the modulations of MAM integrity by NO in a mouse liver are independent of variations of expression of some key proteins at the MAM interface, suggesting that post-transcriptional regulations of MAM proteins should be rather involved. In agreement with the literature [2,49,50], these data suggest that the phosphorylation status of key proteins at the MAM interface may contribute to the regulation of MAM integrity. Of course, we cannot exclude other types of post-transcriptional modifications. These observations are important for understanding regulatory mechanisms involved in the control of MAM integrity, and further works are needed to identify the targeted proteins.

Physiological concentrations of NO are expected to regulate the insulin response in the liver [9] through phosphorylation [17] and S-nitrosylation [18]. The involvement of the sGC/PKG pathway in mediating Akt activation by NO has been described in retinal neurons in culture [17], but the mechanism had not yet been clearly reported in the liver. Our findings therefore add significant evidence that physiological concentrations of NO enhance the hepatic insulin response through the canonical sGC/PKG pathway. More groundbreaking, our findings in vitro propose that MAMs could be a crucial hub for controlling the hepatic response to insulin by NO. Firstly, we exclude that MAM regulation by NO is consecutive to its action on insulin signaling, as we evidenced that MAMs still adapt to NO in the presence of wortmannin, a PI3K/Akt inhibitor. Therefore, the effects of NO on MAMs precede those on insulin signaling, suggesting a potential causative role. In addition, we found no modification of both Akt and GSK3β levels, or their phosphorylation states, in hepatic MAM fractions from treated mice, suggesting that the effect of NO on insulin signaling may be independent of a targeting of insulin signaling proteins at the MAM interface. In order to confirm the involvement of MAMs in NO-regulated insulin signaling, we used two experimental strategies used until now to disrupt ER–mitochondria interactions. It is noteworthy that no optimal method exists to reduce ER–mitochondria interactions as no proteins of MAM are specific of this compartment. Therefore, their modulation could have indirect consequences due to their presence outside MAMs. However, the silencing of different MAM proteins was previously shown to dampen ER–mitochondria interactions, and in general, parallel silencing of several MAM proteins is used to confirm the involvement of MAMs [2,4]. An alternative is to overexpress the spacer FATE-1 [35], as this protein is expressed only in the testis and not in the liver. We previously used this strategy to dampen MAMs in skeletal muscle and confirmed their involvement in muscle insulin resistance [25]. In this study, we decided to combine both experimental strategies, as we disrupted ER–mitochondria interactions either by silencing CypD, a mitochondrial protein interacting with the VDAC1-Grp75-IP3R complex and regulating calcium exchange [24,34], or by overexpressing the protein FATE-1 [25]. Firstly, Huh7 silenced for CypD evidenced disrupted MAMs and lack of positive NO-induced adaptation for both MAM integrity and insulin response. The non-adaptation of MAMs in CypD-KO models may be due to the side effect of the CypD-KO on the cell’s ability to produce NO (Appendix A). However, NONOate also failed to induce positive adaptation of MAM in models of CypD downregulation using either CRIPR/Cas9 (Appendix A) or siRNA (Appendix A), suggesting that CypD could potentially be a target protein regulating the integrity of MAM in response to NO. In addition, disruption of MAMs by FATE-1 blunted the impact of NO on insulin signaling, causing insulin resistance. Altogether, our data support that MAM integrity is required to mediate NO-induced activation of the insulin signaling pathway in the liver.

The link between MAMs and insulin sensitivity is actually highly controversial, whatever the tissue considered. Indeed, hepatic insulin resistance was associated with disrupted [2,24] or enhanced [4,51] ER–mitochondria interactions. In skeletal muscle, the same controversy exists as reduced [25] or excessive [52] MAMs were found in diet-induced obese and diabetic models. Nevertheless, ER–mitochondria miscommunication was confirmed in the kidney [53] and hypothalamus [54] of obese and diabetic mice. The discrepancy between these studies is currently unclear; however, taking into account that several proteins of insulin signaling are present at the MAM interface [1], a shared concept supports that MAMs are a crucial relay platform in the insulin signaling pathway, contributing to its improvement or degradation. Unfortunately, the present study does not give answers to this controversy but points that impaired NO production could lead to ER–mitochondria miscommunication and exacerbate hepatic insulin resistance.

To conclude under physiological conditions, NO is involved through the sGC/PKG pathway in the regulation of MAM integrity, which may participate to the impact of NO on the hepatic insulin response. These observations indicate that NO and the MAM platform could be at the very heart of hepatic physiology. Understanding the regulatory mechanisms involved in the control of MAM integrity and identifying key target proteins is now crucial to identify new therapeutic strategies for preventing or treating hepatic metabolic disorders.

## Figures and Tables

**Figure 1 cells-08-01319-f001:**
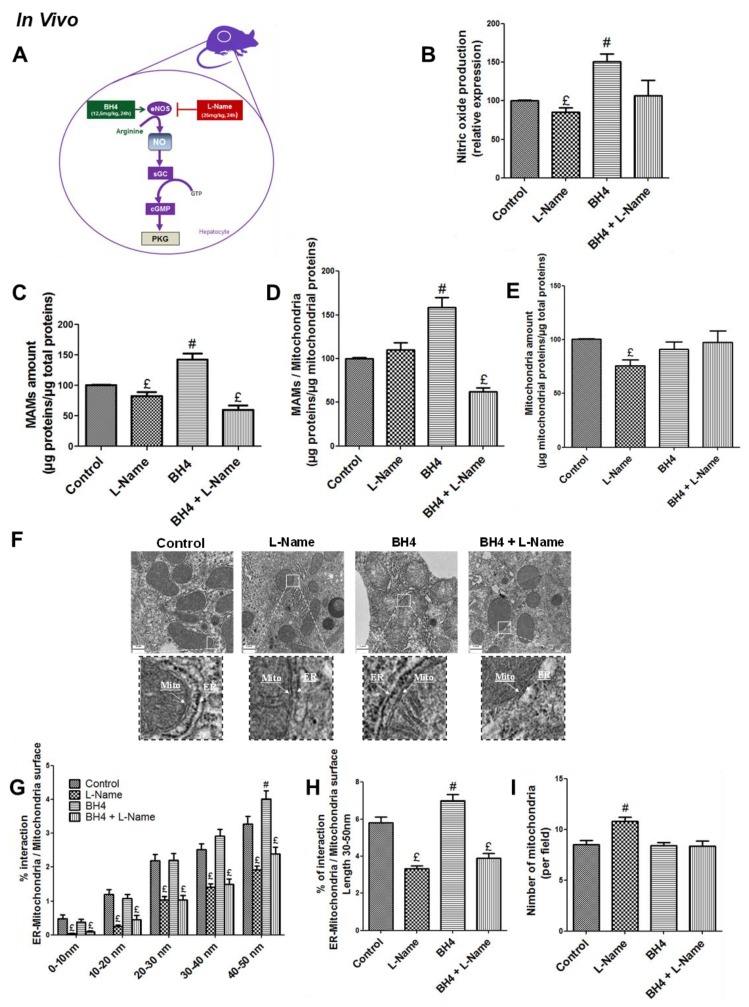
Nitric oxide (NO) regulates mitochondria-associated membranes’ (MAMs) integrity in vivo. (**A**) Representative scheme of strategies used to modulate NO concentration in C57Bl6/JOlash male mice. BH4 (tetrahydrobiopterin; a cofactor of endothelial NO synthase (eNOS), 12.5 mg/kg), l-Name (ω-nitro-l-arginine methyl ester hydrochloride; an inhibitor of eNOS, 25 mg/kg), and a combination of the two were given by intraperitoneal (ip) injection twice the day before and 2 h before euthanasia (*n* = 5 to 7 mice per condition). Impact of modulating in vivo NO production on (**B**) liver NO concentration assessed using Daf-FM (15 µM) on fresh homogenates; (**C**,**D**) amount of proteins in the MAM fraction isolated from fresh liver using differential ultracentrifugation expressed relative to (**C**) total and (**D**) mitochondrial proteins; (**E**) amount of proteins in the mitochondrial fraction isolated from fresh liver using differential ultracentrifugation expressed relative to total proteins; (**F**) endoplasmic reticulum (ER)–mitochondria interactions at different distances (from 0 to 50 nm) analyzed from transmission electronic microscopy (TEM) images. (F) Representative TEM images; (**G**) quantitative analysis of the interactions according to spacing (0–10 nm, 10–20 nm, 20–30 nm, 30–40 nm, 40–50 nm; (**H**) interactions in a 30–50 nm range; (**I**) and the number of mitochondria per field (**I**). A minimum of 10 images (scale bar 0.5 µm) was taken for each mouse (*n* = 160–260 ER–mitochondria interaction analyzed/mice group condition). £ and #, *p* < 0.05 vs. control.

**Figure 2 cells-08-01319-f002:**
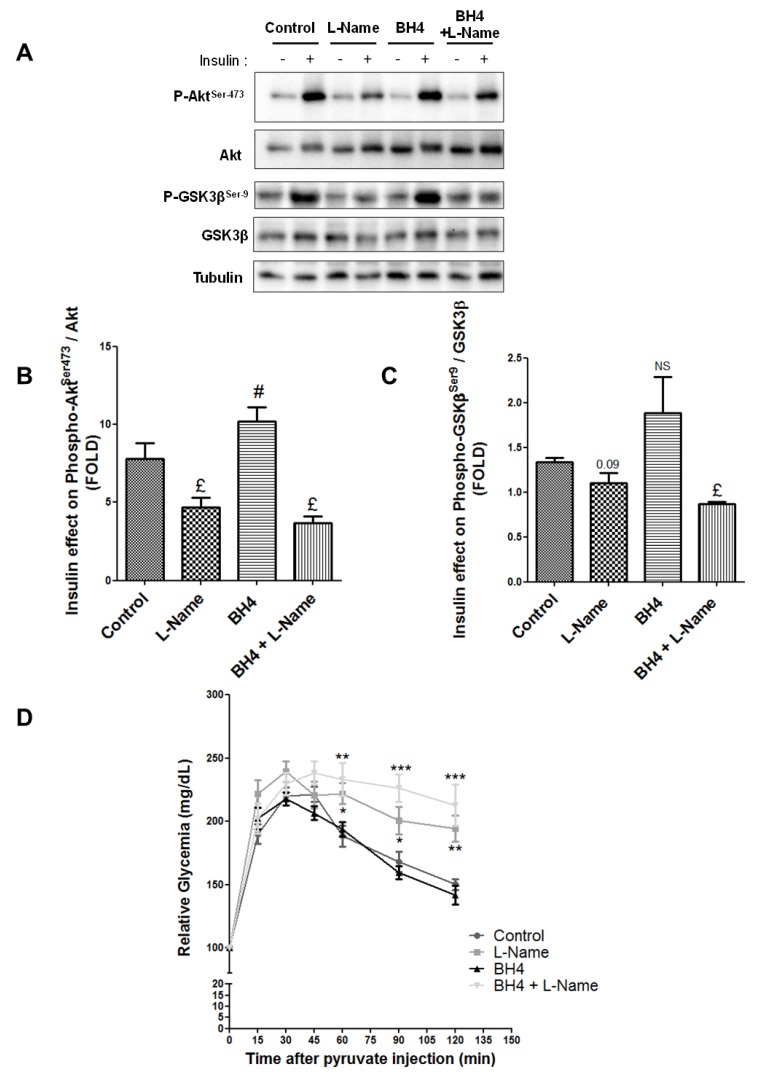
NO regulates the hepatic insulin sensitivity in vivo**.** (**A**–**C**) Impact of modulating in vivo the NO production on hepatic insulin signaling. C57Bl6/JOlash male mice (*n* = 5 mice per condition) received ip injection of BH4 (12.5 mg/kg), l-Name (25 mg/kg), and a combination of the two, twice the day before and 2 h before the insulin test (0.75 U/kg). (**A**) Representative Western blots and quantitative analysis of insulin-stimulated (**B**) protein kinase B (Akt) and (**C**) glycogen synthase kinase 3 beta (GSK3β). (**D**) Impact of modulating in vivo the NO production on hepatic glucose production. C57Bl6/JOlash male mice (*n* = 8 mice per condition) received ip injection of BH4 (12.5 mg/kg), l-Name (25 mg/kg), and a combination of the two, twice the day before and 2 h before the pyruvate test (2 g/kg). £ and #, *p* < 0.05 vs. control. **p* < 0.05, ***p* < 0.01, and ****p* < 0.001 vs. appropriate control glycaemia, respectively.

**Figure 3 cells-08-01319-f003:**
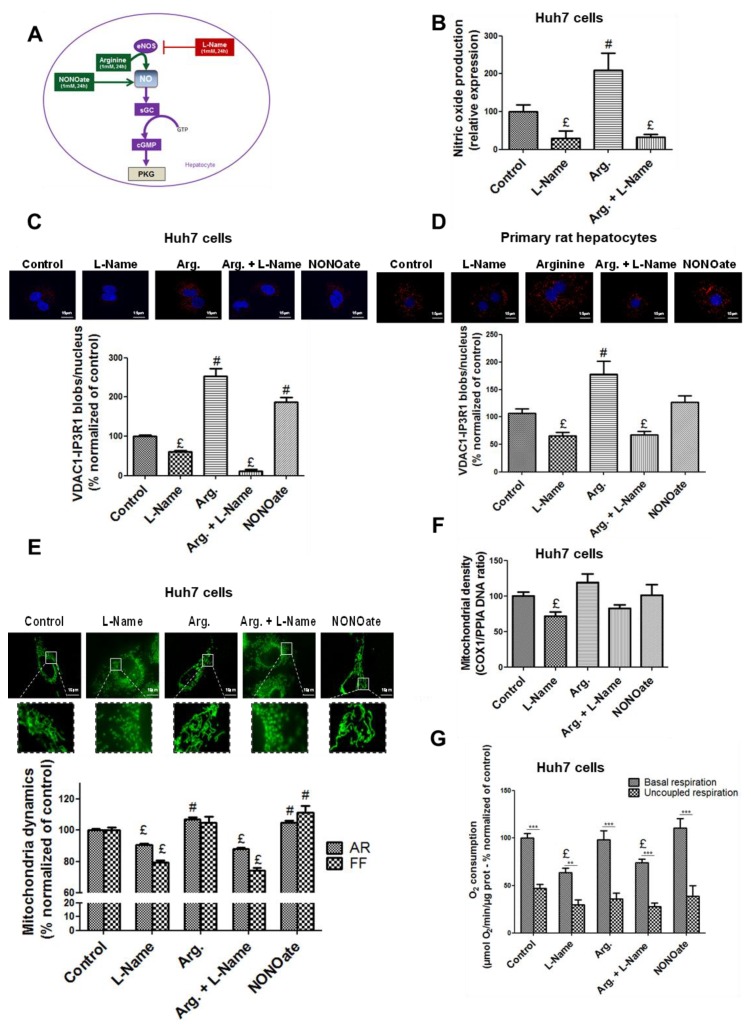
NO regulates the ER–mitochondria contact points and the mitochondrial network architecture, density, and respiratory activity in hepatocytes in vitro. (**A**) Representative scheme of strategies used to modulate NO concentration in vitro. Arginine is a substrate of eNOS, l-Name in an inhibitor of eNOS, and NONOate (diethylamine NONOate sodium salt hydrate) is a NO donor. (**B**) Impact of l-Name (1 mM), arginine (1 mM), and a combination of the two on NO concentration in fresh Huh7 cells assessed using Daf-FM (15 µM). (**C**–**G**) Impact of modulating NO concentration for 24 h using l-Name (1 mM), arginine (1 mM), a combination of the two, and NONOate (1 mM), on (**C**,**D**) ER–mitochondria interactions assessed through VDAC1/IP3R1 (voltage dependent anion channel 1/inositol 1,4,5-trisphosphate receptor) interactions using in situ proximity ligation assay (PLA) in (**C**) Huh7 cells and (**D**) primary rat hepatocytes (*n* = 10 images minimum per experiment, three independent series per treatment, representative image at top and enlarged in Appendix A, and quantitative analysis below, scale bar 15 µm, ×100); (**E**) mitochondrial network architecture, assessed in Huh7 cells using MitoTracker^®^ Green (500 nM) and calculation of aspect ratio (AR) and form factor (FF) (*n* = 10 images minimum per experiment, three independent series per treatment, representative image at top and enlarged in Appendix A, scale bar 15 µm, ×100); (**F**) mitochondrial (COX-1, cyclooxygenase-1) DNA content relative to nuclear (PPIA, peptidylprolyl isomerase A) DNA assessed in Huh7 cells using RT-qPCR (*n* = three independent experiments); (**G**) whole cell oxygen consumption assessed using oxygraphy in Huh7 cells (*n* = minimum six independent experiments). £ and ^#^, *p* < 0.05 vs. control; ***, *p* < 0.05 vs. respective uncoupled state.

**Figure 4 cells-08-01319-f004:**
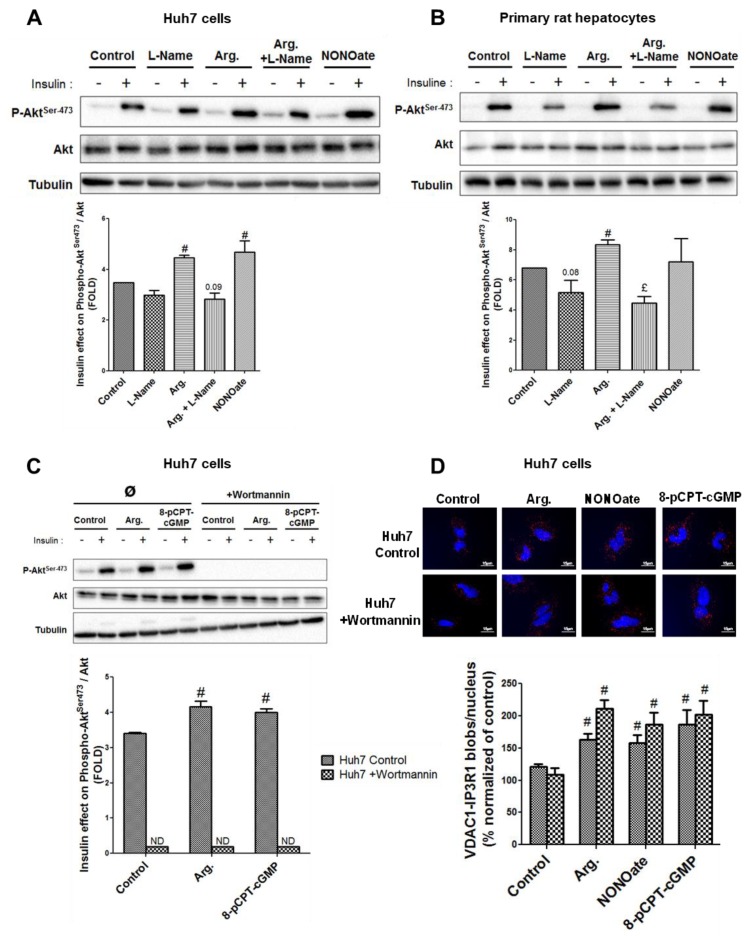
NO regulates the insulin response in hepatocytes in vitro. (**A**,**B**) Impact of modulating NO concentration for 24 h using l-Name (1 mM), arginine (1 mM), a combination of the two, and NONOate (1 mM) on insulin signaling in (**A**) Huh7 cells and (**B**) primary rat hepatocytes. (**C**,**D**) Impact of inhibiting phosphoinositide 3-kinase (PI3K) using wortmannin (1 µM) on the regulation by NO of (**C**) ER–mitochondria interactions assessed using in situ PLA and (**D**) insulin signaling. For Western blot, representative image (at top) and quantitative analysis (below) of insulin-stimulated Akt (*n* = three independent experiments). For in situ PLA, representative image (at top and enlarged in Appendix A) and quantitative analysis of VDAC1/IP3R1 interactions (below) (*n* = 10 images minimum per experiment, three independent series per treatment, scale bar 15 µm, ×100). £ and ^#^, *p* < 0.05 vs. control. £ and #, *p* < 0.05 vs. control.

**Figure 5 cells-08-01319-f005:**
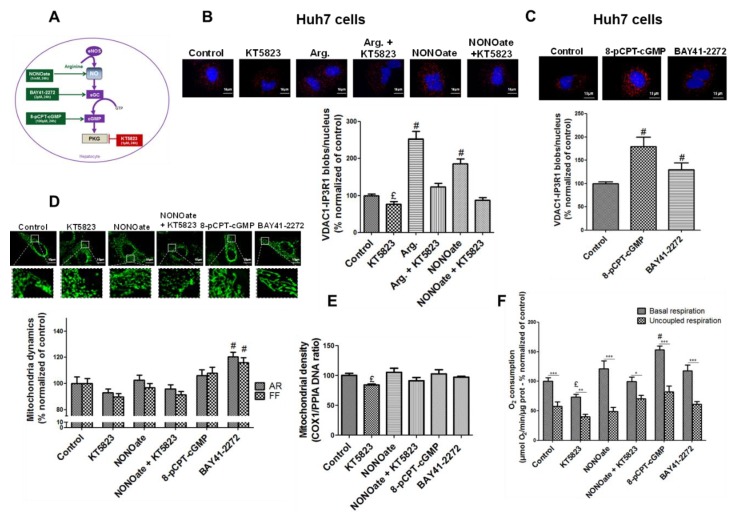
NO-activated subunits of guanylate cyclase/protein kinase G (sGC/PKG) pathway regulates the ER–mitochondria contact points and participates in the modulation of the mitochondrial network architecture, density, and respiratory activity in hepatocytes in vitro. (**A**) Representative scheme of strategies used to modulate the sGC/PKG pathway in vitro. Arginine is the substrate of eNOS, NONOate is a NO donor, BAY41-2272 (3-(4-amino-5-cyclopropylpyrimidin-2-yl)-1-(2-fluorobenzyl)-1*H*-pyrazolo [3–b]pyridine) is an activator of the sGC, 8-pCPT-cGMP (8-(4-chlorophenylthio)-guanosine 3′,5′-cyclic monophosphate sodium salt) activates PKG whereas KT5823 inhibits the kinase. (**B**) Impact of inhibiting the sGC/PKG pathway for 24 h in Huh7 cells using KT5823 (1 µM) in the presence or absense of arginine (1 mM) or NONOate (1 mM), on ER–mitochondria interactions assessed using in situ PLA (*n* = 10 images minimum per experiment, three independent series per treatment, representative images at top and enlarged in Appendix A, and quantitative analysis below, scale bar 15 µm, ×100). (**C**) Impact of activating the sGC/PKG pathway for 24 h in Huh7 cells using 8-pCPT-cGMP (100 µM) and BAY41-2272 (2 µM) on ER–mitochondria interactions assessed using in situ PLA (*n* = 10 images minimum per experiment, three independent series per treatment, representative images at top and enlarged in Appendix A, and quantitative analysis below, scale bar 15 µm, ×100). (**D**–**F**) Impact of modulating the sGC/PKG pathway for 24 h in Huh7 cells using KT5823 (1 µM), NONOate (1 mM), a combination of the two, 8-pCPT-cGMP (100 µM), and BAY41-2272 (2 µM) on (D) mitochondrial network architecture assessed using MitoTracker^®^ Green (500 nM) and calculation of aspect ratio (AR) and form factor (FF) (*n* = 10 images minimum per experiment, three independent series per treatment, representative image at top and enlarged in Appendix A, scale bar 15 µm ×100); (**E**) mitochondrial (COX-1) DNA content relative to nuclear (PPIA) DNA assessed using RT-qPCR (*n* = three independent experiments); (**F**) whole cell oxygen consumption assessed using oxygraphy (*n* = minimum six independent experiments). £ and #, *p* < 0.05 vs. control; ***, *p* < 0.05 vs. respective uncoupled state.

**Figure 6 cells-08-01319-f006:**
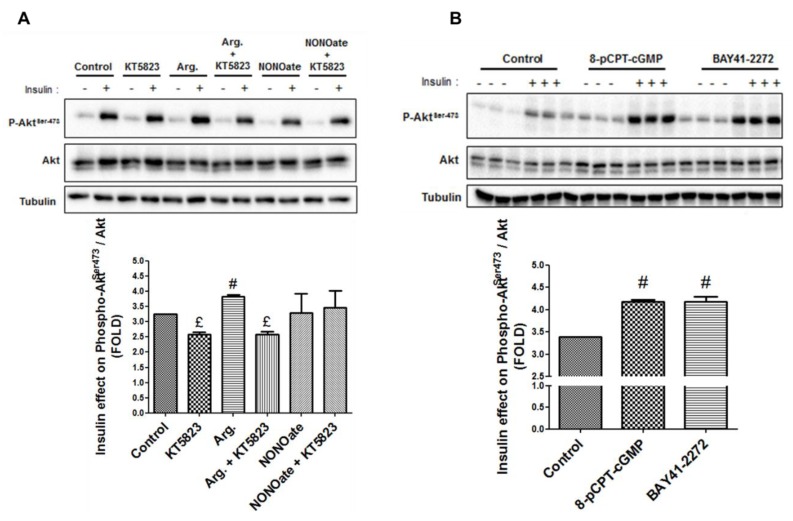
NO-activated sGC/PKG pathway regulates the insulin response in hepatocytes in vitro. Impact on insulin signaling in Huh7 cells (**A**) inhibiting the sGC/PKG pathway using KT5823 (1 µM) in the presence or absence of arginine (1 mM) or NONOate (1 mM) for 24 h; and (**B**) activating the sGC/PKG pathway using 8-pCPT-cGMP (100 µM) and BAY41-2272 (2 µM) for 24 h. ^£^ and ^#^, *p* < 0.05 vs. control.

**Figure 7 cells-08-01319-f007:**
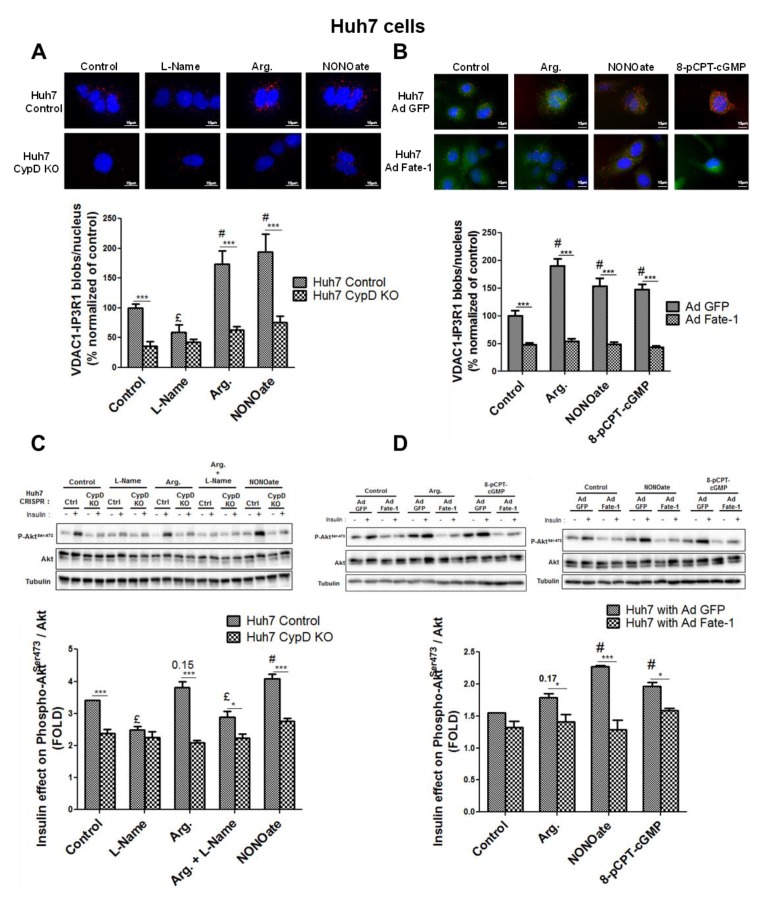
MAM integrity is required for mediating NO impact on hepatic insulin sensitivity in vitro. Impact of modulating NO concentration using l-Name (1 mM), arginine (1 mM), and NONOate (1 mM), in control and CypD-knockout (KO) Huh7 cells on (**A**) ER–mitochondria interactions assessed through VDAC1/IP3R1 interactions using in situ PLA and (**C**) insulin signaling. Impact of modulating NO concentration using arginine (1 mM), NONOate (1 mM), and 8-pCPT-cGMP (100 µM) in Huh7 cells transfected with a control adenovirus (green fluorescent protein; ad GFP) and FATE-1 (fetal and adult testis-expressed 1; ad FATE-1) on (**B**) ER–mitochondria interactions assessed through VDAC1/IP3R1 interactions using in situ PLA and (**D**) insulin signaling. For in situ PLA, representative image (at top and enlarged in Appendix A) and quantitative analysis of VDAC1/IP3R1 interactions (below) (*n* = 10 images minimum per experiment, three independent series per treatment, scale bar 15 µm, ×63 or ×100). For Western blot, representative images (at top) and quantitative analysis (below) of insulin-stimulated Akt (*n* = three independent experiments). ^£^ and ^#^, *p* < 0.05 vs. control. *, ** and ***, *p* < 0.05, *p* < 0.01, and *p* < 0.001 vs. appropriate Huh7 control, respectively.

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
