# Peer review of "Regulation of Mitochondria-Associated Membranes (MAMs) by NO/sGC/PKG Participates in the Control of Hepatic Insulin Response"

_cells, 2019, doi:10.3390/cells8111319_

Round 1

Reviewer 1 Report

cells-605476; Bassot A et al.
Regulation of mitochondria-associated membranes (MAMs) by NO/sCG/PKG participates in the control of hepatic insulin response

In this manuscript, the authors investigate whether NO produced by eNOS upregulates hepatic insulin sensitivity by targeting MAMs using Huh7 cells, primary rat hepatocytes and mouse livers as experimental systems. The results suggested that the induction of MAMs participate to the impact of NO on hepatocyte insulin response.
There are many data which support the conclusion and logic of the paper look good. I agree that the present results might contribute to the related field.
My comments (minor) are as follows.
1. CG may be GC? If so, there are many typos through the whole manuscript.
2. In contrast to Western blots, immunofluoresence data (including Fig. 3C&D, Fig. 4D, Fig. 5B, C&D) are too small to see for me.
3. Are there any types of diseases relevant to insulin resistance in liver?
4. Are there any signaling molecules in the insulin receptor pathway that may interact with MAM?

Author Response

In this manuscript, the authors investigate whether NO produced by eNOS upregulates hepatic insulin sensitivity by targeting MAMs using Huh7 cells, primary rat hepatocytes and mouse livers as experimental systems. The results suggested that the induction of MAMs participate to the impact of NO on hepatocyte insulin response.

There are many data which support the conclusion and logic of the paper look good. I agree that the present results might contribute to the related field.
My comments (minor) are as follows.

CG may be GC? If so, there are many typos through the whole manuscript.

Corrections have been made throughout the manuscript.

In contrast to Western blots, immunofluoresence data (including Fig. 3C&D, Fig. 4D, Fig. 5B, C&D) are too small to see for me.

Because figures are dense and do not allow us to increase the size of the representative images of immunofluorescence, we have chosen to provide these images in large size in the supplementary file. Please see Figure S7.

Are there any types of diseases relevant to insulin resistance in liver?

Liver insulin resistance does play a central role in the pathophysiology of the metabolic syndrome (review by Meshkani R et al. Clin Biochem. 2009;42 :1331-46). In addition, there are close links between liver insulin resistance and extensive metabolic cross-talk with adipose tissue, pancreas and skeletal muscles, leading to a vicious circle (reviewed in Czech MP. Nat Med. 2017;23:804-814). Finally, as reviewed by Perry et al. (Nature. 2014;510:84-91), liver insulin resistance is common to non-alcoholic fatty liver diseases (NAFLD) and type 2 diabetes. Almost all patients with NAFLD have hepatic insulin resistance, which increases the risk of impaired fasting glycemia and type 2 diabetes.

We thus mentioned this important information at the end of the first paragraph of the introduction : « Thus, understanding key mechanisms regulating MAM integrity in the liver could help identifying new strategies to modulate hepatic insulin sensitivity which is a key lever to prevent metabolic disorders associated with obesity [6,7]. » 

Are there any signaling molecules in the insulin receptor pathway that may interact with MAM?

Indeed, from a molecular point of view, several actors involved in the insulin signalling pathway, such as Akt, GSK3b, PP2A, PTEN and mTORC2, have been identified in enriched fractions from MAMs of different cells or tissues (Wang et al., Nature 2012:485:128–132; Gomez et al., Cell Death Differ. 2016;23:313-22; Tubbs et al., Diabetes 2014: 63:3279-94). These proteins are mainly involved in regulating calcium transfer between the two organelles by modulating the activity of the IP3R channel, but also seem to be involved in regulating insulin response (Wang et al., Nature 2012:485:128–132). In particular, data from our team showed an increase in Akt phosphorylation at MAMs following insulin stimulation in liver cells (Tubbs et al., 2014). However, the exact mechanisms linking these actors to the regulation of the insulin signaling pathway are still unknown. To complete our work, we explored two actors of the insulin signalling pathway (Akt -total and P-Ser-473- and GSK3β -total and P-Ser-9-) at the MAMs’ fractions of mice liver. Our results did not evidence significant differences of these proteins, or their phosphorylation state, following modulation of NO levels. These data have been added to the supplementary file. Please see Figure S2A.

We mentioned the results in the first paragraph of the results’ section :

« Some insulin signaling proteins, such as Akt and GSK-3β, are located at MAMs [2] and may be involved in mediating the impact of NO on the insulin response. We thus explored whether NO could alter levels of total and/or phosphorylated Akt and GSK-3β in MAM fractions from mice liver, following acute BH4 and/or L-Name treatments. However, no significant difference was observed between treatments (Fig. S3A). »

Reviewer 2 Report

Remarks to the Author:
In the manuscript titled 'Regulation of mitochondria-associated membranes (MAMs) by NO/sCG/PKG participates in the control of hepatic insulin response’, A. Bassot et al. provides evidence for linking the ER-mitochondrial contact site with NO and hepatic insulin response. The authors observed the change of ER and mitochondrial contact site and mitochondrial network architecture by eNOS inhibitor or arginine addition in vivo and in vitro, which affect the insulin response confirmed by Akt phosphorylation and mediated by the downstream sGC/PKG pathway.

I find that the claims made in this manuscript would be more convincing if the following major and minor points are addressed:

Do the ER-mitochondrial contact sites decrease in eNOS knockout mice liver or eNOS down-regulated hepatic cells? Fig1 C-E, I do not find any particular reasons why authors did not show blots for specific proteins for each fraction and quantify based on the blots. The font size of labels on graphs is quite small and hard to read. Images in Fig 3 C, D, E, Fig 4 D, Fig 5 B, C, D, Fig 7 A, B need to be enlarged. And Fig 5-7 are particularly small and impossible to see PLA dots in the representative images. It is unclear the link between the change of mitochondrial network architecture and the MAM by NO or NO-mediated sCG/PKG pathway. The molecular mechanism of how the NO changes MAM integrity is pretty unclear. The MAM proteins increase upon BH4 addition Fig 1 C-E, however, it is unclear whether the proteins involved in MAM formation accumulated at the MAM or expression levels of MAM proteins increased thus increased the contact sites between these two organelles.

Author Response

Answers to comments from Reviewer 2 :

In the manuscript titled 'Regulation of mitochondria-associated membranes (MAMs) by NO/sCG/PKG participates in the control of hepatic insulin response’, A. Bassot et al. provides evidence for linking the ER-mitochondrial contact site with NO and hepatic insulin response. The authors observed the change of ER and mitochondrial contact site and mitochondrial network architecture by eNOS inhibitor or arginine addition in vivo and in vitro, which affect the insulin response confirmed by Akt phosphorylation and mediated by the downstream sGC/PKG pathway.

I find that the claims made in this manuscript would be more convincing if the following major and minor points are addressed:

Do the ER-mitochondrial contact sites decrease in eNOS knockout mice liver or eNOS down-regulated hepatic cells?

We thank the reviewer for this interesting proposition. We have tried to answer it using the siRNA approach in HuH7 cells. Unfortunately, our initial results questioned the efficacy of the commercial mix of siRNA purchased from Qiagen. We would need more time to explore other siRNAs to complete this approach.

Please see attached file for the illustration of the first results.

Fig1 C-E, I do not find any particular reasons why authors did not show blots for specific proteins for each fraction and quantify based on the blots.

As suggested by the reviewer, we performed Western blotting analysis of specific proteins in the different fractions purified from treated HuH7 cells and quantify their modulation following the modulation of NO. These proteins have been chosen on one hand to validate the subcellular fractionation (Fig. S2A) and on the other hand to differentially analyse the expression of most known MAM proteins in hepatic MAM fractions from treated mice (Fig.S2B). These information have been added to the manuscript (lines 254-275) and illustrated in Fig. S2A and B.

The font size of labels on graphs is quite small and hard to read.

Figures have been improved.

Images in Fig 3 C, D, E, Fig 4 D, Fig 5 B, C, D, Fig 7 A, B need to be enlarged. And Fig 5-7 are particularly small and impossible to see PLA dots in the representative images.

Because figures are dense and do not allow us increasing the size of the representative images of immunofluorescence, we have chosen to provide these images in large size in the supplementary file. Please see Figure S7

It is unclear the link between the change of mitochondrial network architecture and the MAM by NO or NO-mediated sCG/PKG pathway.

The question whether the effect of NO on the mitochondrial network dynamics involves the modulation of MAMs is very interesting and represents a full research objective in its own right. In this study, we have chosen to explore the mitochondrial network dynamics on the HuH7 model only to validate the effects of NO. 

The molecular mechanism of how the NO changes MAM integrity is pretty unclear.

We fully agree with the reviewer’s comment. Understanding the mechanisms of how NO changes MAM integrity is part of our future objectives. As this work requires time, and our data are still preliminary, it is not conceivable to include it to the current manuscript. However, potential mechanisms are suggested in the discussion.

The MAM proteins increase upon BH4 addition Fig 1 C-E, however, it is unclear whether the proteins involved in MAM formation accumulated at the MAM or expression levels of MAM proteins increased thus increased the contact sites between these two organelles.

To answer this question, we explored the expression of key proteins involved in MAM formation. These data, presented in Figure S2B, evidenced no significant impact of treatments on their protein content at the hepatic MAM fractions from treated mice, suggesting that MAM integrity can be modulated without changing the expression of key actors. However, this does not exclude that regulation of other structuring proteins may be involved.

Reviewer 3 Report

In the current manuscript authored by Bassot et al., explores the role of NO in the regulation of ER-mitochondria contact sites and impact on the hepatic insulin signaling. The authors have provided evidence on the impact of NO on the MAM integrity and NO impact on the insulin signaling. The authors have rigorously shown that eNOS produced NO impacts the MAM integrity and insulin signaling separately but I am not completely convinced with the idea of NO induced changes on the MAM integrity affects the insulin signaling. CypD KO or FATE1 over-expression both of which affected the contact sites, also affected the insulin signaling/response. Also NO is affects the mitochondrial function/morphology it might be a secondary or two parallel events downstream of NO. Hence, based on the current format, I recommend the manuscript for major revisions to consider towards publication.

Comments:

Fig S1A & S1B: Need a positive western blot control for the iNOS. May be different tissue sample that expresses iNOS. Fig 1C-D: quantifying the fractions is not the correct method for showing enrichment of proteins at MAM or assessment of MAM integrity. Need Western blots showing the purity of the fractions? And also showing the enrichment of known MAM localized proteins upon NO inducing conditions. Even though it is already known that insulin signaling protein complexes localizes to MAMs, it might be a good idea to be show their presence on MAMs in the current manuscript upon NO inducing/inhibiting treatments. Fig 1B & 1F: BH4 treatment shows only a moderate increase in contacts as compared to the increase in NO upon treatments. Please provide better contrast images for the representative TEM images. Also, please include a plot for difference in the number of mitochondria either per field or per cell upon treatments. Fig 2B: Similar to previous comment, based on the impact of BH4 on Akt activation or NO levels, the same is not reflected (to similar levels) in pyruvate tolerance test showing only moderate or no insulin sensitivity (glycemic response) than control. Please explain further on the variations. Throughout the manuscript the fluorescence images are hard to visualize (PLA dots or mitochondrial network). Please provide better images. Fig S2B: Need cell viability assay (Flow-based or microplate-based assay) in addition to caspase3 blot to look at apoptosis. Fig S2C: It’s interesting to note that ER stress markers (p-EIF2a or p-PERK) are very highly expressed even at basal untreated conditions. Please comment on the same. The authors need to compare the levels of NO between WT vs CypD KO and Control vs FATE-1. Authors might try to constitutively tether the ER and Mitochondrion using artificial tethers (dual membrane localizing) and then look at insulin response upon NO treatments. Minor changes: Please maintain consistency with respect to L-NAME and sGC throughout the manuscript.

Author Response

Answers to comments from reviewer 3

In the current manuscript authored by Bassot et al., explores the role of NO in the regulation of ER-mitochondria contact sites and impact on the hepatic insulin signaling. The authors have provided evidence on the impact of NO on the MAM integrity and NO impact on the insulin signaling. The authors have rigorously shown that eNOS produced NO impacts the MAM integrity and insulin signaling separately but I am not completely convinced with the idea of NO induced changes on the MAM integrity affects the insulin signaling. CypD KO or FATE1 over-expression both of which affected the contact sites, also affected the insulin signaling/response. Also NO is affects the mitochondrial function/morphology it might be a secondary or two parallel events downstream of NO. Hence, based on the current format, I recommend the manuscript for major revisions to consider towards publication.

We understand the reluctance of the reviewer concerning the causal link between NO-regulated MAM integrity and NO-regulated insulin signalling. However, as pointed in our discussion, there is currently no optimal method to validate the involvement of MAMs in a biological effect. We are conscious that modulating CypD or FATE1 is associated per se with disruption of insulin signalling (which is observed in respective control conditions). What we demonstrated further here is that both experimental alteration of MAMs blunted NO effects on insulin signalling. 

Related to the comment on mitochondria morphology, we agree with the reviewer that we did not demonstrate whether MAM modulation by NO was involved in the regulation of mitochondria dynamics. Whereas this question is interesting, it is beyond the scope of the present study.

Comments:

Fig S1A & S1B: Need a positive western blot control for the iNOS. May be different tissue sample that expresses iNOS.

The figures have been corrected accordingly. Liver homogenate from a high fat-fed mouse was used as positive control for iNOS. Please see Fig. S1A and B.

Fig 1C-D: quantifying the fractions is not the correct method for showing enrichment of proteins at MAM or assessment of MAM integrity. Need Western blots showing the purity of the fractions? And also showing the enrichment of known MAM localized proteins upon NO inducing conditions.

We agree with the reviewer that quantifying proteins in MAM fractions is not sufficient to analyse MAM integrity. It is why, we associate to this approach either TEM analysis in mice liver or PLA in HuH7 cells. Nevertheless, as requested by the reviewer, we added in the revised version of the manuscript Western blot analysis to either validate the purity of hepatic MAM fractions (Fig. S2A), or to access differential protein expression of specific MAM proteins in hepatic MAM fractions from treated mice (Fig.S2B). Results from figure S2A validate the purity of subcellular fractionation. Results from figure S2B demonstrate that the expression of key MAM proteins is not modified by modulation of NO in hepatic MAM fractions from treated mice. These points are now mentioned in the revised version of the manuscript.

Even though it is already known that insulin signaling protein complexes localizes to MAMs, it might be a good idea to be show their presence on MAMs in the current manuscript upon NO inducing/inhibiting treatments.

Indeed, from a molecular point of view, several actors involved in the insulin signalling pathway, such as Akt, GSK3b, PP2A, PTEN and mTORC2, have been identified in enriched fractions from MAMs of different cells or tissues (Wang et al., Nature 2012:485:128–132; Gomez et al., Cell Death Differ. 2016;23:313-22; Tubbs et al., Diabetes 2014: 63:3279-94). These proteins are mainly involved in regulating calcium transfer between the two organelles by modulating the activity of the IP3R channel, but also seem to be involved in regulating insulin response (Wang et al., Nature 2012:485:128–132). In particular, data from our team showed an increase in Akt phosphorylation at MAMs following insulin stimulation in liver cells (Tubbs et al., 2014). However, the exact mechanisms linking these actors to the regulation of the insulin signaling pathway are still unknown. To complete our work, we explored two actors of the insulin signalling pathway (Akt -total and P-Ser-473- and GSK3β -total and P-Ser-9-) at the MAMs’ fractions of mice liver. Our results did not evidence significant differences of these proteins, or their phosphorylation state, following modulation of NO levels. These data have been added to the supplementary file. Please see Figure S2A.

We mentioned the results in the first paragraph of the results’ section :

« Some insulin signaling proteins, such as Akt and GSK-3β, are located at MAMs [2] and may be involved in mediating the impact of NO on the insulin response. We thus explored whether NO could alter levels of total and/or phosphorylated Akt and GSK-3β in MAM fractions from mice liver, following acute BH4 and/or L-Name treatments. However, no significant difference was observed between treatments (Fig. S3A). »

Fig 1B & 1F: BH4 treatment shows only a moderate increase in contacts as compared to the increase in NO upon treatments. Please provide better contrast images for the representative TEM images.

TEM images have been improved accordingly. Please see Fig. 1F.

Also, please include a plot for difference in the number of mitochondria either per field or per cell upon treatments.

We thank the reviewer for the proposition. Information regarding the number of mitochondria per field of TEM images from mice liver has been added to Fig 1I. Whereas the number of mitochondria per field was similar in the liver between control, BH4 and BH4 + L-Name treatments, data showed an increase in the number of mitochondria per field in response to L-Name. These data suggest that inhibiting eNOS modulate a process of mitochondria life, such as mitochondriogenesis, mitochondria dynamics, and/or mitophagy. Determining the cause of this increase requires further investigations which are beyond the scope of this study.

Fig 2B: Similar to previous comment, based on the impact of BH4 on Akt activation or NO levels, the same is not reflected (to similar levels) in pyruvate tolerance test showing only moderate or no insulin sensitivity (glycemic response) than control. Please explain further on the variations.

We agree with the reviewer that data from PTT are not so convincing than those on the impact of treatments on insulin signalling in the liver. We think that this difference is related to the fact that systemic assessment of insulin sensitivity is less sensitive as influenced by other physiological parameters of mice, whereas analysis of insulin-stimulated Akt in the liver is more robust. In addition, as treatments are performed in healthy mice (under standard diet), we suspected that it is more difficult to observe an improvement of hepatic insulin sensitivity despite increase of insulin signalling.

Throughout the manuscript the fluorescence images are hard to visualize (PLA dots or mitochondrial network). Please provide better images.

Because figures are dense and do not allow us to increase the size of the representative images of immunofluorescence, we have chosen to provide these images in large size in the supplementary file. Please see Figures S7.

Fig S2B: Need cell viability assay (Flow-based or microplate-based assay) in addition to caspase3 blot to look at apoptosis.

Thank you for the comment. Two distinct measures of cell viability (Thoma cell counting chamber, and automatic counting using the LUNATM Automated Cell Counter, Logos Biosystems) have been added to Fig. S4B. They showed similar cell viability between treatments.

The information is mentioned in the results’ section: « Finally, we found no effect of NO modulation on cell viability (Fig. S4B) and on the expression of caspase 3 and ER stress markers, excluding any effects on mitochondrial apoptosis (Fig. S4C) and on ER stress (Fig. S4D). »

Fig S2C: It’s interesting to note that ER stress markers (p-EIF2a or p-PERK) are very highly expressed even at basal untreated conditions. Please comment on the same.

Our analyses did not show any difference between treatments in terms of markers of ER stress. The high intensity signal is due to the fact that the antibodies are not of good quality, that we used LuminataTM forte and that we chose images with a clear signal. Of course, we cannot exclude that it reflects small stress in our culture conditions.

The authors need to compare the levels of NO between WT vs CypD KO and Control vs FATE-1.

Thanks for the constructive comment. Initially, we did not perform this experiment as we used both Ad-GFP and Ad-FATE1-IRES-GFP which both overexpress GFP and the measurement of NO is based on green fluorescence (Daf-FM signal). Therefore, we used here two other adenoviruses, Ad-cherry in control and Ad-FATE1 (without GFP) and measured NO production in infected HuH7 cells.  NO production in Ad Fate-1 vs. Ad Cherry infected cells have been added to Fig. S3E. These measurements show that cells transfected with ad FATE1 are still responsive to modulation of NO production, since L-Name decreases the NO level and arginine enhances it. In addition, NO levels in CypD KO vs. CRISPR control cells are now compared on the same graph. Please, see Fig. S3D. This new presentation shows, as mentioned in the manuscript, that CypD KO cells are characterized by an alteration in the ability to produce NO. However, treatment with NONOate also did not modify MAMs or improve insulin response, arguing that disruption of MAM integrity by cypD KO blunts the impact of NO on the insulin response.

Authors might try to constitutively tether the ER and Mitochondrion using artificial tethers (dual membrane localizing) and then look at insulin response upon NO treatments.

Thank you for this proposal. Such an approach would indeed be interesting. However, for the moment, we are not in a position to test it because the artificial tethers have not been validated in our model.

 Minor changes: Please maintain consistency with respect to L-NAME and sGC throughout the manuscript.

Corrections have been made throughout the manuscript.

Round 2

Reviewer 2 Report

Fig1 C-E, I do not find any particular reasons why authors did not show blots for specific proteins for each fraction and quantify based on the blots.

As suggested by the reviewer, we performed Western blotting analysis of specific proteins in the different fractions purified from treated HuH7 cells and quantify their modulation following the modulation of NO. These proteins have been chosen on one hand to validate the subcellular fractionation (Fig. S2A) and on the other hand to differentially analyse the expression of most known MAM proteins in hepatic MAM fractions from treated mice (Fig.S2B). These information have been added to the manuscript (lines 254-275) and illustrated in Fig. S2A and B.

The newly provided data Fig S2A and B are very strange and terrible performed. First, authors need to choose the proteins and antibodies that total homogenates express all proteins, and second, why coomassie staining for loading control with the samples from liver, which the easiest tissues to obtain proteins, and MAMs fractions, and third, Opa1 is not known to be in MAMs fraction so far, it is at the inner mitochondrial membrane facing to the intermembrane space. Besides, other proteins tested, Mfn2, Grp75, CypD, VDAC1 are all mitochondrial proteins, not ER. Thus, this MAMs fraction is not MAMs fraction, it is mitochondrial fraction.

Reviewer 3 Report

The authors have made suggested changes to the manuscript which has improved the quality of the manuscript. 

I recommend the manuscript for publication in Cells journal in the current format.